# SPREPE: A SPHERICAL GEOMETRY-AWARE POSITION EMBEDDING FOR VISION TRANSFORMERS

## ABSTRACT

Position embedding (PE) is a key mechanism that breaks the permutation symmetry of tokens in Transformer, introducing a spatial inductive bias that enables attention to model locality, distances, and directional relations. Spherical data arise in many scientific domains, most notably in astronomy and meteorology, where Vision Transformers is increasingly adopted for the ability to capture long-range dependencies. However, conventional PEs are designed for linear sequences and cannot faithfully capture the sphere's non-Euclidean geometry. Furthermore, existing designs for encoding spherical positional information rely on additional network modules or specialized network architectures, which introduce extra parameters and computational overhead. These limitations motivate a geometry-aware and efficient embedding scheme that fully exploits spherical structure to advance Transformer-based modeling on the sphere. We introduce **Spherical Reflection Position Embedding (SpRePE)**, a lightweight method efficiently leveraging spherical positional information for Vision Transformer. SpRePE encodes the absolute position on the sphere using a Householder matrix and incorporates the explicit relative position dependency into the attention formulation, achieving both high computational efficiency and high accuracy without requiring substantial additional parameters and modifications to the overall model architecture. We evaluate SpRePE on representative tasks, including spherical image classification and global weather forecasting. SpRePE consistently outperforms well-known baselines including APE, RPE, ALiBi and RoPE. These results indicate that SpRePE offers an efficient and broadly applicable position embedding scheme for Transformer models on the sphere.

## 1 INTRODUCTION

The Transformer architecture (Vaswani et al., 2017) was first introduced for natural language processing tasks and has since demonstrated a strong non-linear modeling capacity across numerous benchmarks (Vaswani et al., 2017; Radford et al., 2019; Devlin et al., 2019). Its vision counterpart, Vision Transformer (ViT), applies the same attention principle to sequences of image patches and achieves competitive or superior performance compared to convolutional neural networks, particularly for modeling long-range dependencies (Dosovitskiy et al., 2021; Liu et al., 2021).

Since the attention mechanism itself is position-agnostic, a Transformer model without position embedding (PE) treats all tokens as interchangeable, severely limiting representational ability (Yun et al., 2020). Injecting positional information is therefore indispensable for capturing token order or spatial layout, and preserving accuracy for Transformer (Vaswani et al., 2017; Shaw et al., 2018; Devlin et al., 2019; Lan et al., 2020) as well as Vision Transformer (Dosovitskiy et al., 2021; Heo et al., 2024).

Spherical data are widely used across many scientific domains, including geoscience (Alken et al., 2021), astronomy (Planck Collaboration et al., 2020), and meteorology (Hersbach et al., 2018). In recent years, Transformer-based models have also been introduced to address problems involving spherical data, with notable examples including Pangu-Weather (Bi et al., 2023) for weather forecasting and WenHai (Cui et al., 2025) for oceanography.

Unlike planar Cartesian coordinates, the spherical domain is non-Euclidean. Spherical coordinates obey periodic boundary conditions in longitude and exhibit coordinate singularities at the poles.

These properties pose specific challenges for positional encoding when applying Vision Transformers on the sphere.

Prior works have explored various PE strategies. One class relies on *absolute* position embeddings (APE), which can be either fixed (Vaswani et al., 2017) or learnable (Devlin et al., 2019; Lan et al., 2020; Clark et al., 2020). These schemes add positional information at the stem to break permutation symmetry. However, APE does not directly encode relative positional information and provides little geometry-aware inductive bias; such relations must be learned implicitly. A second class employs *relative* position embedding (RPE), which provides positional information based on token-to-token offsets and encodes it into the attention mechanism (Shaw et al., 2018; Huang et al., 2020; Ke et al., 2021; He et al., 2021; Press et al., 2022; Su et al., 2024). These schemes are mainly designed for planar grids; therefore, they may fail to faithfully represent geodesic relations under spherical periodicity and singularity. Moreover, several spherical PE methods incorporate positional information via custom operators (Yun et al., 2023) or additional neural modules (Mai et al., 2023), which increases model size and training overhead.

To address the above challenges and the shortcomings of current PEs, we introduce SpRePE, a reflection-matrix-based spherical position embedding scheme for Vision Transformer. SpRePE encodes absolute positions on the sphere using reflection matrices and directly injects an explicit spherical relative-position term into the attention mechanism. The reflection-matrix formulation endows this relative term with a clear geometric interpretation, thereby respecting spherical geometric constraints. SpRePE adopts a drop-in design that operates solely within the attention module, requiring neither substantial additional parameters nor modifications to the overall model architecture. In practice, the extra computation overhead is negligible compared with the attention computation.

To validate the effectiveness of SpRePE, we conduct experiments on representative spherical tasks: spherical image classification and global weather forecasting. The experiments are implemented within the same Vision Transformer framework to ensure fair and consistent evaluation. We compare SpRePE with well-known position embedding methods including APE, RPE, ALiBi and RoPE. The results show that SpRePE performs strongly across tasks, with particularly pronounced advantages when conventional position embedding are affected by geometric distortions. These results demonstrate that SpRePE is an efficient and practical position embedding method for attention-based networks in spherical modeling tasks.

## 2 PRELIMINARY

Consider a standard self-attention layer with input sequence $\mathbf{X} = [x_1, x_2, \ldots, x_n]^\top \in \mathbb{R}^{n \times d}$, where $n$ is the number of tokens and $d$ is the hidden dimension. The attention output is computed as:

$$f(\mathbf{X}) = \text{Attention}(\mathbf{X}) = \text{softmax}\left(\frac{\mathbf{Q}\mathbf{K}^\top}{\sqrt{d}}\right)\mathbf{V},$$

where $\mathbf{W}^{\mathbf{t}:\mathbf{t} \in \{\mathbf{Q}, \mathbf{K}, \mathbf{V}\}} \in \mathbb{R}^{d \times d}$ are parameter matrices, $\mathbf{Q} = \mathbf{X}\mathbf{W}^\mathbf{Q}$, $\mathbf{K} = \mathbf{X}\mathbf{W}^\mathbf{K}$, and $\mathbf{V} = \mathbf{X}\mathbf{W}^\mathbf{V}$.

Without position embedding, the self-attention layer is permutation-equivariant, meaning for any permutation $\sigma \in S_n$, we have

$$f(P_\sigma \mathbf{X}) = P_\sigma f(\mathbf{X}),$$

where $P_\sigma$ is the permutation matrix corresponding to $\sigma$. This symmetry makes it impossible for the model to distinguish positions of tokens.

**Absolute position embedding (APE) & Relative position embedding (RPE)**   To break this symmetry, two families of position embedding schemes are widely adopted: absolute position embedding (APE) and relative position embedding (RPE). Implementation details of both families are deferred to the Appendix B. Given their practicality and prevalence, we adopt representative implementations from each family as baselines in our experiments.

**Rotary position embedding (RoPE)**   Although RPE incorporates relative positional information, its direct application to the attention mechanism limits its influence on $q$ and $k$, thereby constraining its effectiveness. Additionally, compare to APE, RPE incurs significant higher computational

cost. Motivated by these limitations, RoPE encodes positional information by applying rotations to $q$ and $k$, such that their inner product depends only on relative offsets, while introducing negligible overhead relative to standard attention. Specifically, in the one-dimensional case, let the $n$-th query and the $m$-th key be denoted by $q_n, k_m \in \mathbb{R}^d$. RoPE provides a solution to the constraint $\langle f(q_n, n), f(k_m, m) \rangle = g(q_n, k_m, n - m)$, i.e. the inner product after position embedding depends only on the relative offset $n - m$.

RoPE is defined as $\text{RoPE}(q_n, n) = R_n \circ \hat{q}_n$, where $\circ$ denotes element-wise multiplication. The complex vector $\hat{q}_n \in \mathbb{C}^{d/2}$ is obtained by pairing consecutive real dimensions:

$$\text{Re}(\hat{q}_{n,t}) = q_{n,2t}, \text{Im}(\hat{q}_{n,t}) = q_{n,2t+1}, t = 0, \dots, \frac{d}{2} - 1.$$

The rotary factor $R_n \in \mathbb{C}^{d/2}$ is given by $R_{n,t} = e^{in\theta_t}$, with fixed frequencies $\theta_t$ independent of $n$.

In natural language processing, RoPE has been shown to perform well and to excel at length extrapolation. Two extensions are commonly adopted in Vision Transformers:

**(i) Axial–frequency RoPE.** This extension keeps each coordinate is encoded independently:

$$R_{n,2t} = e^{i\,x_n\theta_t}, R_{n,2t+1} = e^{i\,y_n\theta_t}, \quad t = 0, \dots, \frac{d}{4} - 1.$$

The axial variant is a natural extension of 1D RoPE to two dimensions: it applies independent 1D RoPE along the height and width axes to span a 2D grid. This design is simple to implement, maintains a bijection between rotary phase and spatial location, and has become the de facto standard; we therefore adopt it as the state-of-the-art baseline in our experiments.

**(ii) Mixed–frequency RoPE.** For a token located at $p_n = (x_n, y_n)$, the rotary factor is defined as

$$R_{n,t} = \exp\left\{ i\left(x_n\theta_t^x + y_n\theta_t^y\right) \right\}, \qquad t = 0, \dots, \frac{d}{2} - 1.$$

This formulation blends the horizontal and vertical offsets, enabling the model to consider both directions simultaneously. However, like the axial variant, it operates under a planar parameterization and does not endow any sphere-specific properties.

All relative position embeddings above model displacement solely through index differences. While this assumption is adequate on a planar grid, it fails on a sphere due to the non–Euclidean geometry:

- **Equator vs. pole.** Under a Plate Carrée projection, a longitudinal difference of $180°$ spans half the globe at the equator but collapses to an infinitesimal arc near the poles.

- **Prime–meridian wrap–around.** Near the prime meridian, two points with the maximal longitudinal index difference appear distant in the projection, yet are almost coincident on the sphere.

Hence, index–based relative position embeddings cannot faithfully capture spherical distances, motivating our geometry–aware positional embedding for spherical data.

## 3 METHOD

We propose **Spherical Reflection Position embedding (SpRePE)**, a spherical position embedding scheme, in which *absolute* embedding results inform *relative* information through the inner product of encoded queries and keys. In other words, the embedding algorithm only requires the position of a single token at embedding time, and allows the attention mechanism to reflect the pairwise relative positional information between any two tokens.

More concretely, when a Vision Transformer processes spherical data, let $p_1 = (\theta_{p_1}, \phi_{p_1})$ and $p_2 = (\theta_{p_2}, \phi_{p_2})$ be two arbitrary points on the unit sphere, where $\theta \in [-\pi, \pi]$ and $\phi \in [0, 2\pi)$ denote longitude and latitude, respectively. SpRePE provides a solution $f(\cdot, \cdot)$ such that, for query vector $q$ and key vector $k$,

Table 1: Comparison of position embedding schemes.

| Property | APE | RPE | RoPE | SpRePE |
|---|---|---|---|---|
| Where applied | stem stage | attention score | $q/k$ vectors | $q/k$ vectors |
| Compute-efficient | ✓ | ✗ | ✓ | ✓ |
| Relative info | ✗ | ✓ | ✓ | ✓ |
| Sphere-aware | ✗ | ✗ | ✗ | ✓ |

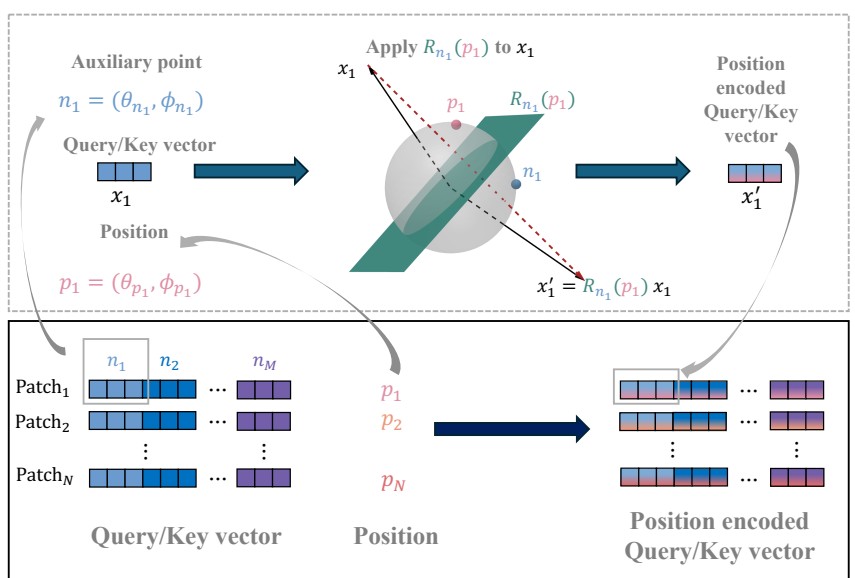

Figure 1: **Implementation of Spherical Reflection Position embedding (SpRePE)**. The embedding space is partitioned into $d/3$ disjoint 3D subspaces, each processed independently by SpRePE. For each subspace, we preselect an auxiliary point $n_m = (\theta_{n_m}, \phi_{n_m})$ on the unit sphere. Given a patch located at $p_i = (\theta_{p_i}, \phi_{p_i})$, the Householder reflection matrix $R_{n_m}(p_i)$ is applied to the corresponding query/key vector $x_i$, yielding the position-encoded vector $x_i' = R_{n_m}(p_i) x_i$. This scheme ensures that the embedding stage relies solely on the *absolute* coordinates of $p_i$. The upper part of the figure illustrates the special case with $m = 1$, $i = 1$.

$$\langle f(q, p_1), f(k, p_2) \rangle = g\big(q, k, \text{pos}(p_1, p_2)\big), \tag{1}$$

where $\text{pos}(p_1, p_2)$ denotes a function of the *relative* position between $p_1$ and $p_2$ on the sphere. Table1 compares SpRePE with the well-known PE schemes through several key characteristics. Its RoPE-like design enables the incorporation of spherical relative positional information while keeping the computational overhead negligible.

## 3.1 DERIVATION OF SPREPE UNDER 3D

We begin with a special case with a dimensions $d = 3$. Because the admissible choices of the pair $(f, g)$ satisfying Eq.1 admit large degrees of freedom, we have ample room to impose additional structure. We first convert the spherical coordinates to Cartesian form for convenience, $p_1 = \big(\sin\theta_{p_1}\cos\phi_{p_1}, \sin\theta_{p_1}\sin\phi_{p_1}, \cos\theta_{p_1}\big)$, $p_2 = \big(\sin\theta_{p_2}\cos\phi_{p_2}, \sin\theta_{p_2}\sin\phi_{p_2}, \cos\theta_{p_2}\big)$. We may conveniently impose the constraint that $f$ acts as multiplication by an orthogonal matrix; that is, $f(q, n) = R(n)\, q$. Even under this constraint, $f$ remains sufficiently flexible.

Note that

$$g\big(q, k, \text{pos}(p_1, p_2)\big) = \langle f(q, p_1), f(k, p_2) \rangle = q^\top R(p_1)^\top R(p_2)\, k = q^\top R(p_1, p_2)\, k \tag{2}$$

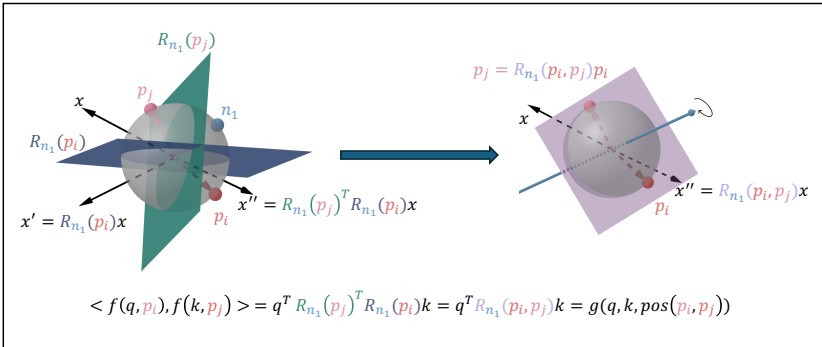

Figure 2: **Illustration of how SPREPE encodes *relative* position information through *absolute* reflections and an inner-product ($d = 3$).** The Householder matrix is applied to the query/key vector $x_i$, yielding the position-encoded vector $x_i' = R_{n_1}(p_i) x_i$. Given two patches $p_i$ and $p_j$, the product $R_{n_1}(p_i)^\top R_{n_1}(p_j)$ is a rotation whose axis equals the normal of the plane defined by $n$, $p_1$, and $p_2$. Consequently, the inner product $\langle f(q, p_1), f(k, p_2) \rangle = q^\top R_n(p_1)^\top R_n(p_2) k$ depends only on $q$, $k$, and the relative position $\mathrm{pos}(p_1, p_2)$, thereby realizing Eq.1 without ever observing both coordinates simultaneously.

where $R(p_1, p_2) = R(p_1)^\top R(p_2)$. Because both $R(p_1)$ and $R(p_2)$ are orthogonal with $\det R(p_1) = \det R(p_2)$, we have $\det R(p_1, p_2) = \det R(p_1) \det R(p_2) = 1$. Hence, $R(p_1, p_2)$ is itself an orthogonal matrix with unit determinant, i.e. a three-dimensional rotation.

We impose

$$p_1^\top R(p_1, p_2) \, p_2 \;=\; p_1^\top R(p_1)^\top R(p_2) \, p_2 \;=\; p_1^\top p_1 \;=\; p_2^\top p_2 \;=\; 1. \tag{3}$$

Accordingly, $R(p_1, p_2)$ is the rotation matrix that maps $p_2$ onto $p_1$. The matrix $R(p_1, p_2)$ still carries one degree of freedom—its rotation angle $\gamma$. Once the angle is chosen, with $\gamma \in \left[\arccos\langle p_1, p_2\rangle, \pi\right]$, the rotation axis can be chosen from at most two alternatives.

However, we cannot parameterize $R(p_1, p_2)$ by its rotation angle, because evaluating that angle requires simultaneous knowledge of both $p_1$ and $p_2$, whereas the embedding scheme has access to only one of the two points at a time. Consequently, the rotation angle is not a suitable independent variable.

A more reasonable choice is to introduce an auxiliary point $n$. For arbitrary point $p$ on the sphere, let $R_n(p)$ denote the Householder reflection that mirrors $p$ onto $n$. that is, the unique orthogonal matrix satisfying $R_n(p) \, n = p$ and $R_n(p)^\top = R_n(p)^{-1}$ with $\det R_n(p) = -1$. This matrix serves as the fundamental building block of our spherical position embedding.

Once $n$ is fixed, the matrix $R(p_1, p_2)$ becomes fully determined. Notice that the normal vector of the plane defined by $n$, $p_1$, and $p_2$ is exactly the eigenvector of both $R_n(p_1)$ and $R_n(p_2)$, and therefore serves as the rotation axis of the matrix $R_n(p_1)^\top R_n(p_2)$. A more complete proof is provided in the Appendix C. Given this axis, the rotation angle is uniquely defined as the ratio of arc length between $p_1$ and $p_2$ along the circle obtained by intersecting that plane with the sphere to its radius.

The degree of freedom associated with $n$ is therefore redundant: any two choices of $n$ that generate the same plane yield identical embedding. We exploit this observation to avoid double-counting during embedding. Because the grid is fixed *a priori*, we simply enforce that, for every pair $(p_1, p_2)$, any two auxiliary points $n_1$ and $n_2$ must define distinct sections; Under this constraint, distinct choices of $n$ produce distinct embedding results.

It is worth emphasizing that the proposed embedding scheme imposes no assumptions on the grid: it is equally applicable to cubed-sphere as well as latitude–longitude discretizations. Likewise, it places no restrictions on the backbone architecture; the only requirement is the presence of an attention mechanism.

We next derive the explicit form of $R_n(p_1)$. Let $p_1$, $n$ be two unit vectors on the sphere. The Householder reflection that maps $p_1$ onto $n$ is $R_n(p_1) = I - 2vv^\top, v = \frac{n - p_1}{\|n - p_1\|} = (v_1, v_2, v_3)^\top$.

## 3.2 GENERAL FORM

In the general setting where the embedding dimension is a multiple of three, we partition the d-dimensional space into $M = d/3$ three–dimensional subspaces and apply an independent reflection to each. Formally, the spherical position embedding acts as

$$f(x, p) = R(p) x, R(p) = \text{diag}(R_{n_1}(p), R_{n_2}(p), \dots, R_{n_M}(p)). \tag{4}$$

Each $R_{n_i}(p)$ is the Householder matrix that mirrors $p$ onto the auxiliary point $n_i$.

Distinct choices of the auxiliary points $\{n_i\}_{i=1}^{M}$ therefore yield a family of block–diagonal reflection matrices that act independently on disjoint triplets of feature dimensions.

Taking the advantage of the sparsity of $R(p)$ in Eq. 4, we can avoid computing the matrix–vector product between $R(p)$ and $x \in \mathbb{R}^d$; instead, we use an element-wise multiplication for a more efficient computation. Specifically, the form is

$R(p)x =$

$$\begin{bmatrix} 1 - 2v_{(n_1,p)_1}^2 \\ 1 - 2v_{(n_1,p)_2}^2 \\ 1 - 2v_{(n_1,p)_3}^2 \\ 1 - 2v_{(n_2,p)_1}^2 \\ 1 - 2v_{(n_2,p)_2}^2 \\ 1 - 2v_{(n_2,p)_3}^2 \\ \vdots \\ 1 - 2v_{(n_M,p)_1}^2 \\ 1 - 2v_{(n_M,p)_2}^2 \\ 1 - 2v_{(n_M,p)_3}^2 \end{bmatrix} \circ \begin{bmatrix} x_1 \\ x_2 \\ x_3 \\ x_4 \\ x_5 \\ x_6 \\ \vdots \\ x_{d-2} \\ x_{d-1} \\ x_d \end{bmatrix} - \begin{bmatrix} 2v_{(n_1,p)_1} v_{(n_1,p)_2} \\ 2v_{(n_1,p)_2} v_{(n_1,p)_3} \\ 2v_{(n_1,p)_3} v_{(n_1,p)_1} \\ 2v_{(n_2,p)_1} v_{(n_2,p)_2} \\ 2v_{(n_2,p)_2} v_{(n_2,p)_3} \\ 2v_{(n_2,p)_3} v_{(n_2,p)_1} \\ \vdots \\ 2v_{(n_M,p)_1} v_{(n_M,p)_2} \\ 2v_{(n_M,p)_2} v_{(n_M,p)_3} \\ 2v_{(n_M,p)_3} v_{(n_M,p)_1} \end{bmatrix} \circ \begin{bmatrix} x_2 \\ x_3 \\ x_1 \\ x_5 \\ x_6 \\ x_4 \\ \vdots \\ x_{d-1} \\ x_d \\ x_{d-2} \end{bmatrix} - \begin{bmatrix} 2v_{(n_1,p)_1} v_{(n_1,p)_3} \\ 2v_{(n_1,p)_2} v_{(n_1,p)_1} \\ 2v_{(n_1,p)_3} v_{(n_1,p)_2} \\ 2v_{(n_2,p)_1} v_{(n_2,p)_3} \\ 2v_{(n_2,p)_2} v_{(n_2,p)_1} \\ 2v_{(n_2,p)_3} v_{(n_2,p)_2} \\ \vdots \\ 2v_{(n_M,p)_1} v_{(n_M,p)_3} \\ 2v_{(n_M,p)_2} v_{(n_M,p)_1} \\ 2v_{(n_M,p)_3} v_{(n_M,p)_2} \end{bmatrix} \circ \begin{bmatrix} x_3 \\ x_1 \\ x_2 \\ x_6 \\ x_4 \\ x_5 \\ \vdots \\ x_d \\ x_{d-2} \\ x_{d-1} \end{bmatrix}$$

$$\tag{5}$$

where $\circ$ denotes the element-wise multiplication.

## 3.3 REDUCING DENSITY-INDUCED DEGENERACY VIA PARTIAL SPREPE

When position embedding depends solely on token's location on the sphere, non-uniform token distributions may cause *representation degeneracy*: tokens concentrated in certain regions receive highly similar positional features, making $Q/K$ poorly conditioned and reducing the effective rank. To avoid this, we adopt a simple strategy that applies SpRePE only to a subset of the $Q/K$ elements:

$$Q = (1 - M) \circ Q_0 + M \circ \text{SpRePE}(Q_0, p), \quad K = (1 - M) \circ K_0 + M \circ \text{SpRePE}(K_0, p),$$

where $Q_0$ and $K_0$ are the vanilla projections (not identical across patches) and $M \in \{0, 1\}^d$ is a binary mask applied along the hidden dimension.

We consider two mask designs. (i) **Fixed-ratio mask**: SpRePE is applied on a constant fraction $\rho$ of elements (e.g., $\rho = 7/8$), i.e., $M(j) = \mathbb{1}\{j \leq \lfloor \rho d \rfloor\}$. (ii) **Density-adaptive mask**: let $\rho_i \in [\rho_{\min}, \rho_{\max}]$ depend on token $i$'s local sampling density. For equirectangular grids, we use the area weight $a(\phi_i) = \cos \phi_i$ ($\phi_i \in [-\pi/2, \pi/2]$, proportional to solid-angle per token) and set $\rho_i = \text{clip}(\rho_{\max} a(\phi_i), \rho_{\min}, \rho_{\max})$. This gating preserves SpRePE's geometry-aware bias in sparsely sampled regions while preventing over-aggregation in dense areas.

## 3.4 COMPUTATIONAL OVERHEAD

Table 2 compares the extra computation cost introduced by each position embedding scheme. We analyze a generic ViT with $L$ layers, a hidden dimension of $d$, $h$ attention heads, and an input batch of $B$ tensor, each represented by $N = H \times W$ patch tokens. Only leading-order terms are reported; constants and non-dominant factors are omitted for clarity.

Compared to well-known position embedding schemes, RPE introduces substantially higher computational and memory overhead. This is primarily due to its explicit construction of $N \times N$ attention

Table 2: Asymptotic overhead of inference for various position embeddings.

| PE scheme | Time Complexity | Space Complexity |
|---|---|---|
| APE | $\mathcal{O}(BNd)$ | $\mathcal{O}(Nd)$ |
| RPE | $\mathcal{O}(BhN^2L)$ | $\mathcal{O}(hN^2L)$ |
| RoPE | $\mathcal{O}(BNdL)$ | $\mathcal{O}(NdL)$ |
| **SpRePE** | $\mathcal{O}(BNdL)$ | $\mathcal{O}(NdL)$ |

bias matrices for each head in every layer, resulting in both computational and memory complexity that scales quadratically with the sequence length of $N$. Such overhead becomes increasingly prohibitive in high-resolution settings, presenting a critical bottleneck when scaling to high resolution weather forecasting tasks.

In contrast, APE introduces only a one-time overhead during the input embedding stage, with both its time complexity and memory usage growing linearly with the sequence length of $N$. Since the dominant attention operations ($QK^\top$ and $\mathrm{softmax}(\cdot)V$) scale as $O(N^2d/h + Nd^2)$ in time and $O(hN^2 + Nd)$ in memory, this cost is negligible relative to the overall computation of the ViT backbone. Similarly, RoPE and SpRePE apply positional encoding to each token at every layer, with overhead that scales linearly with the sequence length of $N$, hidden dimension $d$, and number of layers $L$. Notably, despite introducing stronger geometric constrain through reflections, SpRePE maintains the same asymptotic complexity as RoPE.

This computational efficiency enables SpRePE to encode relative positional information without relying on costly attention bias matrices. As a result, it provides an effective balance between scalability and expressiveness, making it particularly well-suited for tasks defined on sphere.

## 4 EXPERIMENT

We integrate SpRePE into a standard ViT (Dosovitskiy et al., 2021) and compare performance by replacing only the PE under identical hyperparameter settings. We evaluate SpRePE on representative tasks: spherical image classification on Spherical MNIST (Esteves et al., 2020) and weather forecasting on ERA5 (Hersbach et al., 2018). The baselines include No Position Embedding (NoPE), APE (Dosovitskiy et al., 2021), RPE (Liu et al., 2021), ALiBi (Fuller et al., 2023), and RoPE (Heo et al., 2024). In addition, to assess the effect of our degeneracy-mitigation strategy, we report two collaborative variants: **SpRePE(F Mask)** and **SpRePE(A Mask)**, which respectively use a fixed-ratio mask and a density-adaptive mask. The results show that SpRePE performs remarkable across tasks, highlighting its advantages for spherical data processing.

### 4.1 SPHERICAL IMAGE CLASSIFICATION

We evaluate SpRePE on spherical image classification using the Spherical MNIST dataset, which places handwritten digits near the South Pole to stress correct handling of spherical geometry. Model and training hyper-parameters are provided in the Appendix D. Table 3 reports Top-1 accuracy on the test set, keeping the ViT backbone identical across methods while varying only the PE scheme.

The result shows that SpRePE(F Mask) achieves the highest Top-1 accuracy (96.74%), followed by SpRePE(A Mask) (96.32%) and APE (96.29%). The Fixed-ratio mask variant improves over RoPE by +0.88 pp and over ALiBi by +10.05 pp, under identical backbones and hyper-parameters.

### 4.2 WEATHER FORECASTING

We verify SpRePE in Weather Forecasting on ERA5 dataset (Hersbach et al., 2018). Due to the limitation of resource, we train our models on a subset of atmospheric variables, sub-sampled at a temporal frequency of 6 hours and $128 \times 256$ spatial resolution of the ERA5 dataset. The input sample has 70 channels correspond to the surface and upper-air variables at different vertical layers. The variables used are listed in Appendix D.

Table 3: Spherical MNIST Top-1 accuracy (%) on test set. **Bold** denotes the best score; underline marks the second best. The same convention is used throughout unless otherwise noted.

| NoPE | APE | RPE | ALiBi | RoPE | **SpRePE** | SpRePE(F Mask) | SpRePE(A Mask) |
|------|------|------|-------|------|--------|----------------|----------------|
| 78.81 | 96.29 | 74.05 | 86.69 | 95.86 | 96.04 | **96.74** | 96.32 |

Table 4: Auto-regressive forecasting results on the ERA5 dataset using different position embedding schemes. Performance is evaluated by Anomaly Correlation Coefficient (ACC, higher is better). We measure ACC at the 4th(+24h), 20th (+120h) and 40th (+240h) steps.

| PE scheme | ACC at 24h ↑ | | | ACC at +120h ↑ | | | ACC at +240h ↑ | | |
|-----------|--------|--------|--------|--------|--------|--------|--------|--------|--------|
| | $Z500$ | $T2M$ | $T850$ | $Z500$ | $T2M$ | $T850$ | $Z500$ | $T2M$ | $T850$ |
| NoPE | 0.9931 | 0.9632 | 0.9755 | 0.8199 | 0.8251 | 0.8044 | 0.5286 | 0.6767 | 0.5770 |
| APE | 0.9954 | 0.9746 | 0.9795 | 0.8818 | 0.9087 | 0.8653 | 0.5971 | 0.8119 | 0.6539 |
| RPE | 0.9963 | 0.9659 | 0.9782 | 0.9000 | 0.8560 | 0.8636 | 0.6037 | 0.7260 | 0.6377 |
| ALiBi | 0.9958 | 0.9649 | 0.9780 | 0.8875 | 0.8533 | 0.8535 | 0.5953 | 0.7267 | 0.6398 |
| RoPE | **0.9971** | 0.9683 | 0.9810 | **0.9172** | 0.8741 | 0.8811 | 0.6215 | 0.7502 | 0.6591 |
| **SpRePE** | 0.9964 | 0.9661 | 0.9788 | 0.9089 | 0.8722 | 0.8736 | 0.6144 | 0.7557 | 0.6587 |
| **SpRePE(F Mask)** | 0.9965 | **0.9758** | **0.9813** | 0.9121 | **0.9230** | **0.8886** | **0.6333** | **0.8258** | **0.6783** |
| **SpRePE(A Mask)** | 0.9964 | 0.9754 | 0.9809 | 0.9110 | 0.9226 | 0.8882 | 0.6237 | 0.8168 | 0.6705 |

We use 6 years of ERA5 data (2015-2020): 2015-2018 are used for training, 2019 is used for validation, and 2020 is held out as out-of-sample test set. During training, the network's input is ERA5 fields at time $t$ and is optimized to predict the fields at $t + 6\,\mathrm{h}$, with geometry-weighted L2 Loss. For validation and testing, the model is initialized with the analysis fields at $t = 0$ and then run auto-regressively for 40 steps, producing forecasts at lead times $+6\,\mathrm{h}, +12\,\mathrm{h}, \ldots, +240\,\mathrm{h}$.

Table 4 summarizes auto-regressive forecasting results on key variables, including geopotential at 500hPa (Z500), temperature at 850hPa (T850) and temperature at 2m above surface (T2M). At +24h, RoPE attains the best ACC on Z500, whereas **SpRePE(F Mask)** is highest on T2M and T850; **SpRePE(A Mask)** is the runner-up on both. For long-range skill, at +120h **SpRePE(F Mask)** widens the gap over RoPE on T2M and T850 while remaining slightly lower on Z500; by +240h it surpasses RoPE across all variables, indicating that SpRePE's advantage grows from +24h to +120h and culminates in a complete overtake at +240h.

We hypothesize that spherical geometry has limited impact at the first prediction step, where a planar or grid approximation is adequate, and that its importance increases as the lead time grows. Figure 3 reports the average difference in absolute relative error between RoPE and SpRePE(F Mask) on the test set after 40-step auto-regressive inference; SpRePE(F Mask) exhibits noticeably lower error than RoPE near the pole. For an intuitive visualization of the corresponding attention patterns on the sphere, please refer to Appendix E (Figure 5). By injecting a geometry-aware bias, our method preserves spatial coherence at long lead times and consequently yields higher ACC.

To further assess the generalization of SpRePE, we evaluate robustness to input rotations without retraining: we apply a rigid rotation around the axis normal to the prime-meridian plane. Fig. 4 reports the one-step RMSE. Across all three variables, RMSE increases with the rotation angle. For Z500 and T850, SpRePE maintains lower error than RoPE at all angles shown. For T2m, RoPE is slightly better at very small rotations (up to about $\pi/8$), but SpRePE overtakes it for larger rotations and the gap widens toward $\pi/2$. These results indicate that SpRePE generalizes more robustly under coordinate rotations, consistent with its geometry-aware design.

## 5 RELATED WORK

**Position embeddings for Vision Transformers** In vision tasks, positional information is commonly injected via absolute positional embeddings (APE) or relative positional biases (RPB) (Dosovitskiy et al., 2021; Liu et al., 2021), with lightweight variants such as LaPE (Yu et al., 2023) and ALiBi-style distance decay (Fuller et al., 2023). Rotary positional embeddings (RoPE) (Su et al., 2024) have also been adapted to vision—either as 1D rotations along token order (Jeevan & Sethi,

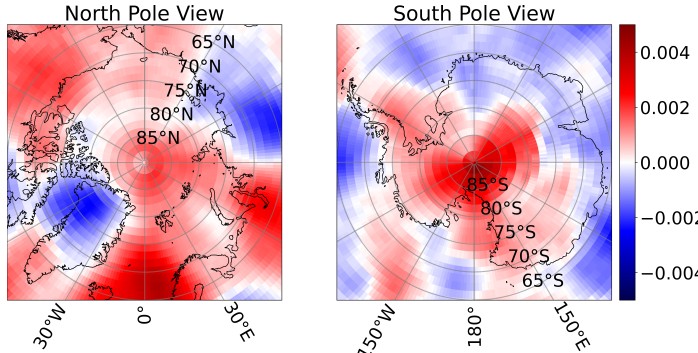

Figure 3: **Polar views of the average difference in absolute relative error between RoPE and SpRePE after 40-step autoregressive inference on the test set.** Each pixel shows $\mathrm{Avg}\big(\mathrm{AbsRel}_{\mathrm{RoPE}} - \mathrm{AbsRel}_{\mathrm{SpRePE}}\big)$, where $\mathrm{AbsRel} = \frac{|y_{\mathrm{pred}} - y_{\mathrm{gt}}|}{|y_{\mathrm{gt}}|}$. Red denotes regions where SpRePE is more accurate (lower AbsRel); blue indicates the opposite.

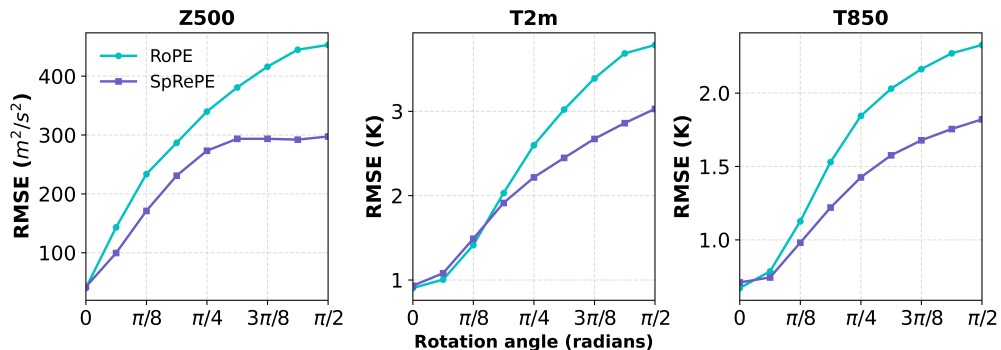

Figure 4: **Robustness to rotated inputs on ERA5.** One-step RMSE versus rotation angle ($0$ to $\pi/2$) on test set. Inputs are rigidly rotated around the axis normal to the prime-meridian plane.

2022) or as 2D extensions applied axially (height/width) or in fused form (Fang et al., 2024; Heo et al., 2024). However, these designs assume Euclidean grids; when applied to equirectangular projections (ERP) of spherical data, they may introduce artifacts near the poles and along the prime-meridian seam (Chen et al., 2023; Lam et al., 2023; Hu et al., 2025).

**Spherical positional representation** Beyond PEs, a separate line of work encodes spherical structure through specialized bases or architectural changes: classical DFS and multi-scale sinusoidal encodings (Orszag, 1974; Mai et al., 2020; 2022), spherical-harmonics embeddings (Rußwurm et al., 2024), and learnable mappers such as Sphere2Vec (Mai et al., 2023). Architecture-level adaptations include explicit latitude/longitude injection or HEALPix partitioning with local RPE (Yun et al., 2023; Carlsson et al., 2024). While effective, these approaches typically introduce custom operators or additional modules, increasing complexity and training cost. Our method, by contrast, remains a *drop-in positional embedding* that operates entirely inside standard attention, requiring no architectural modification.

# 6 CONCLUSION

We present **SpRePE**, a geometry-aware position embedding scheme for spherical data within attention mechanisms. SpRePE encodes absolute positions via reflection matrices and augments attention with an explicit relative-position term. This design only requires that the model include an attention mechanism and imposes no additional requirements on the model architecture. We evaluated SpRePE across multiple tasks, and the results demonstrate its effectiveness and advantages.

**Ethics Statement** This paper studies a positional embedding scheme for Transformers on spherical data. Our experiments use publicly available benchmark datasets; we did not collect new data and processed no personally identifiable information. All datasets are properly cited in the paper. We acknowledge that all authors of this work have read and commit to adhering to the ICLR Code of Ethics.

**Reproducibility statement** We provide an anonymous repository at `https://anonymous.4open.science/r/SpRePE-7357`, which contains (i) the core implementation used in our experiments and (ii) environment specifications required to run them. The repository further includes configuration files, scripts for data preparation, training and evaluation entry points.

Appendix D provides a complete description of the datasets used in the experiments. After downloading the raw data, the datasets can be generated using the scripts we provide in the repository.

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

## A  THE USE OF LARGE LANGUAGE MODELS

According to the conference requirements, we acknowledge the use of large language models during manuscript preparation. Its contributions are translation, polishing, and general writing assistance. Authors take full responsibility for the content of this paper.

## B  IMPLEMENTATION DETAILS OF POSITION EMBEDDINGS

### B.1  ABSOLUTE POSITION EMBEDDING (APE)

Given an input sequence $\mathbf{X} = [x_1, \ldots, x_n] \in \mathbb{R}^{n \times d}$, APE adds a unique positional vector $e_k^{\text{APE}}$ to each token at the stem stage:

$$\widetilde{f}(x_1, \ldots, x_n) = f(x_1 + e_1^{\text{APE}}, \ x_2 + e_2^{\text{APE}}, \ \ldots, \ x_n + e_n^{\text{APE}}).$$

APE is commonly implemented in two ways: *sinusoidal* and *learnable*. For ViT on a $H \times W$ grid, let the $k$-th token have 2D coordinates $(p_k^x, p_k^y)$. The sinusoidal APE of dimension $d$ is defined as

$$
\begin{aligned}
e_k^{\text{APE}}(4t) &= \sin\left(\frac{p_k^x}{10^{\,4t/(\lfloor d \rfloor/4)}}\right), \\
e_k^{\text{APE}}(4t+1) &= \cos\left(\frac{p_k^x}{10^{\,4t/(\lfloor d \rfloor/4)}}\right), \\
e_k^{\text{APE}}(4t+2) &= \sin\left(\frac{p_k^y}{10^{\,4t/(\lfloor d \rfloor/4)}}\right), \\
e_k^{\text{APE}}(4t+3) &= \cos\left(\frac{p_k^y}{10^{\,4t/(\lfloor d \rfloor/4)}}\right),
\end{aligned}
\tag{6}
$$

for $t = 0, 1, \ldots$. Alternatively, $e_k^{\text{APE}} \in \mathbb{R}^d$ can be set as a trainable parameter and learned end-to-end. In our experiments, this learnable APE serves as the APE baseline.

### B.2  RELATIVE POSITION EMBEDDING (RPE)

APE is straightforward and efficient but does not encode relative positions. A common remedy is to add a *relative positional bias* (RPB) directly to the attention logits:

$$\text{Attn}'(i, j) \ = \ \text{Attn}(i, j) \ + \ M_{i,j}^{\text{RPB}},$$

where $M_{i,j}^{\text{RPB}} = e^{\text{RPB}}(\Delta p_{ij})$ and

$$\Delta p_{ij} \ = \ \left(p_i^x - p_j^x, \ p_i^y - p_j^y\right).$$

For a $H \times W$ grid, $e^{\text{RPB}} \in \mathbb{R}^{(2H-1) \times (2W-1)}$ stores a bias value for each possible relative offset. The table $e^{\text{RPB}}$ can be either learnable or predefined (e.g., as a deterministic function of inter-token distance). In all comparisons, we adopt a learnable RPB implementation as the RPE baseline.

## C  GEOMETRIC INTERPRETATION OF SPREPE

In this appendix we make explicit the geometric meaning of SpRePE and explain how it guides the choice of auxiliary points. Recall that for each token position $p_k \in \mathbb{S}^2$ and each auxiliary point $n_\ell \in \mathbb{S}^2$, SpRePE applies a Householder reflection

$$R_{n_\ell}(p_k) \ = \ I - 2\, v_{n_\ell}(p_k)\, v_{n_\ell}(p_k)^\top, \qquad v_{n_\ell}(p_k) \ = \ \frac{n_\ell - p_k}{\big\| n_\ell - p_k \big\|}. \tag{7}$$

The attention inner product between positions $p_i$ and $p_j$ in the subspace associated with $n_\ell$ involves the matrix $R_{n_\ell}(p_i)^\top R_{n_\ell}(p_j)$. We now show that this matrix admits a precise geometric interpretation as a 3D rotation that encodes the relative position between $p_i$ and $p_j$ on the sphere.

Let $n, p_i, p_j \in \mathbb{S}^2$ be three non-collinear points. They determine a unique plane

$$P_{\{p_i, p_j, n\}} = \{ n + \alpha\, v_n(p_i) + \beta\, v_n(p_j) : \alpha, \beta \in \mathbb{R} \}, \tag{8}$$

spanned by $v_n(p_i)$ and $v_n(p_j)$. A unit normal of this plane is given by

$$u = \frac{v_n(p_i) \times v_n(p_j)}{\left\| v_n(p_i) \times v_n(p_j) \right\|_2}, \tag{9}$$

which is well-defined because $v_n(p_i)$ and $v_n(p_j)$ are linearly independent.

We can now state the main geometric result.

**Proposition (Geometric form of the SpRePE encoding).** Let $n, p_i, p_j \in \mathbb{S}^2$ be non-collinear points and define the Householder reflections

$$R_n(p_i) = I - 2\, v_n(p_i)\, v_n(p_i)^\top, \qquad R_n(p_j) = I - 2\, v_n(p_j)\, v_n(p_j)^\top. \tag{10}$$

Then the matrix $R_n(p_i)^\top R_n(p_j)$ satisfies:

1. It is a proper rotation in $\mathbb{R}^3$ (orthogonal with determinant $+1$);

2. Its rotation axis is the line spanned by $u$, i.e., the normal vector to $P_{\{p_i, p_j, n\}}$;

3. Restricted to the circle $P_{\{p_i, p_j, n\}} \cap \mathbb{S}^2$, this rotation maps $p_j$ to $p_i$, and the rotation angle equals the central angle between $p_i$ and $p_j$ along that circle.

**Proof.** We first show that $u$ is a rotation axis of $R_n(p_i)^\top R_n(p_j)$. Since $u$ is orthogonal to both $v_n(p_i)$ and $v_n(p_j)$, we have

$$u^\top v_n(p_i) = u^\top v_n(p_j) = 0. \tag{11}$$

For $m \in \{i, j\}$,

$$R_n(p_m)\, u = \left( I - 2\, v_n(p_m) v_n(p_m)^\top \right) u = u, \tag{12}$$

so $u$ is an eigenvector of both $R_n(p_i)$ and $R_n(p_j)$ with eigenvalue 1, and hence

$$R_n(p_i)^\top R_n(p_j)\, u = R_n(p_i)^\top (R_n(p_j)\, u) = R_n(p_i)^\top u = u. \tag{13}$$

Thus $u$ is an eigenvector of $R_n(p_i)^\top R_n(p_j)$ with eigenvalue 1, so the line spanned by $u$ is a candidate rotation axis.

Each $R_n(p_m)$ is a Householder reflection and therefore an orthogonal matrix with determinant $-1$. Their product $R_n(p_i)^\top R_n(p_j)$ is thus orthogonal with determinant $+1$. An orthogonal $3 \times 3$ matrix with determinant $+1$ and a real eigenvalue 1 must be a proper rotation about the eigenspace associated with eigenvalue 1. This establishes items (1) and (2).

For item (3), note that $R_n(p_j)$ is a reflection across the plane orthogonal to $v_n(p_j)$ and passing through the origin. Geometrically, this reflection swaps $n$ and $p_j$ while preserving the circle $P_{\{p_i, p_j, n\}} \cap \mathbb{S}^2$. Similarly, $R_n(p_i)$ swaps $n$ and $p_i$ and also preserves the same circle. Consequently, the composition $R_n(p_i)^\top R_n(p_j)$ maps $p_j$ to $p_i$ along that circle. Since the axis is orthogonal to $P_{\{p_i, p_j, n\}}$, the restriction of $R_n(p_i)^\top R_n(p_j)$ to $P_{\{p_i, p_j, n\}} \cap \mathbb{S}^2$ is a planar rotation taking $p_j$ to $p_i$, and its angle is exactly the central angle between $p_i$ and $p_j$ on that circle. $\square$

This proposition has two immediate consequences for SpRePE.

First, it provides a clear geometric interpretation of the encoding. For each auxiliary point $n_\ell$, the associated SpRePE subspace effectively represents the relative position between tokens $p_i$ and $p_j$ as a 3D rotation: the rotation axis is the normal of the plane $P_{\{p_i, p_j, n_\ell\}}$ determined by $p_i$, $p_j$, and $n_\ell$, and the rotation angle is the central angle between $p_i$ and $p_j$ along the circle $P_{\{p_i, p_j, n_\ell\}} \cap \mathbb{S}^2$. In this sense, relative positions on the sphere are encoded not by arbitrary linear transforms but by geometrically interpretable rotations.

Table 5: Variables used in our experiments. The "Type" column indicates whether the variable represents a static property, a time-varying single-level property (e.g., surface variables are included), or a time-varying atmospheric property.

| Type | Variable name | Short name | Role |
|---|---|---|---|
| Atmospheric | Geopotential | Z | Input/Predicted |
| Atmospheric | Specific humidity | Q | Input/Predicted |
| Atmospheric | Temperature | T | Input/Predicted |
| Atmospheric | U component of wind | U | Input/Predicted |
| Atmospheric | V component of wind | V | Input/Predicted |
| Single | 2 metre temperature | T2M | Input/Predicted |
| Single | 2 metre Specific humidity | Q2M | Input/Predicted |
| Single | 10 metre u wind component | 10U | Input/Predicted |
| Single | 10 metre v wind component | 10V | Input/Predicted |
| Single | Mean sea level pressure | msl | Input/Predicted |
| Static | Land-sea mask | lsm | Input |
| Static | Geopotential at surface | Z | Input |
| Clock | Zenith Angle | n/a | Input |

Second, the same geometric characterization directly motivates our rule for choosing auxiliary points. If two auxiliary points $n_\ell$ and $n_q$ yield the same plane $P_{\{p_i,p_j,n\}}$ for given $p_i$ and $p_j$, then

$$R_{n_\ell}(p_i)^\top R_{n_\ell}(p_j) \;=\; R_{n_q}(p_i)^\top R_{n_q}(p_j),$$

so the corresponding subspaces carry redundant positional information. Accordingly, our selection scheme is designed so that, given all positions $p$ in advance, each auxiliary point is chosen to ensure that the resulting planes $P_{\{p_i,p_j,n_\ell\}}$ are always distinct, thereby avoiding the above degeneracies. An ablation experiment on the auxiliary-point selection scheme is provided in Appendix F.

## D EXPERIMENT SETTING

**Computing environment.** Experiments are conducted on a single node with $8\times$ NVIDIA A100-PCIe-40GB GPUs. The NVIDIA driver version is 550.54.14 with CUDA 12.4. The host CPU is a dual-socket Intel Xeon Gold 6248R @ 3.0 GHz (24 cores per socket; physical cores / 96 threads in total), organized into two NUMA nodes with AVX/AVX2/AVX-512 support.

**Spherical image classification** Dataset preparation follows (Esteves et al., 2023; 2020). After downloading the original MNIST dataset, the Spherical MNIST variant can be generated using the scripts provided in our repository. We use a standard Vision Transformer (ViT) with patch size $2 \times 2$, length 8, a hidden dimension of 384, and 8 attention heads. Training uses a batch size of 16 and an initial learning rate of $3 \times 10^{-4}$. We optimize with Adam ($\beta_1 = 0.9$, $\beta_2 = 0.999$) and apply a cosine-annealing learning-rate scheduler for 40 epochs.

**Weather Forecasting** Table5 lists all variables used in the experiments. For all atmospheric variables, we use 13 pressure levels: 50, 100, 150, 200, 250, 300, 400, 500, 600, 700, 850, 925, and 1000 hPa.

In experiments, We use a standard Vision Transformer (ViT) with patch size $2\times2$, length 12, a hidden dimension of 384, and 8 attention heads. Training uses a batch size of 4 and an initial learning rate of $5 \times 10^{-4}$. We optimize with Adam ($\beta_1 = 0.9$, $\beta_2 = 0.999$) and apply a cosine-annealing learning-rate scheduler for 20 epochs.

## E ATTENTION VISUALIZATION RESULTS OF SPREPE

Figure 5 further illustrates the geometric effect of SpRePE by visualizing attention maps on the sphere. We consider two extreme query locations: one near the North Pole and one at the intersection

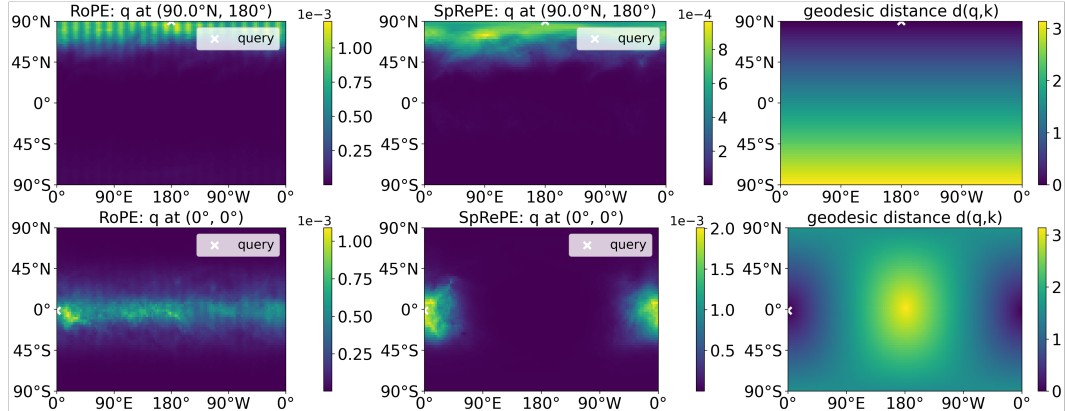

Figure 5: Attention maps of a representative attention head for queries at the North Pole (top) and at the equator–prime-meridian intersection (bottom), together with the corresponding geodesic distance. Across test samples, this head consistently shows that SpRePE aligns attention with spherical geometry more faithfully than RoPE.

of the equator and the prime meridian. When the query lies near the pole, RoPE produces a strip-like attention pattern along the top rows of the ERP grid, whereas SpRePE concentrates attention in a localized region around the pole that decays smoothly toward lower latitudes, consistent with geodesic distance. When the query is placed on the equator at the image boundary, RoPE largely ignores the wrap-around adjacency between the left and right edges, while SpRePE correctly assigns high attention to points that are far apart in the ERP image but adjacent on the sphere. These visualizations provide an intuitive view of the "attention geometry" induced by SpRePE and support the interpretability of its geometry-aware design.

# F  ABLATION STUDIES: AUXILIARY-POINT SELECTION AND MASKING

In this section, we conduct ablation studies on two aspects: the auxiliary-point selection strategy and the masking strategy.

**Auxiliary-point selection.** We compare the following four strategies:

- **Random**: In each layer, auxiliary points are sampled uniformly at random on the sphere and kept fixed;

- **Grid**: In each layer, auxiliary points are chosen from a fixed latitude–longitude grid and kept fixed;

- **Learnable**: In each layer, auxiliary points are initialized randomly and optimized jointly with the model parameters;

- **Identical**: In each layer, all auxiliary points coincide at the same location.

We adopt the same experimental setup as in the main-text experiments on the ERA5 dataset. The results are presented in Table 6.

The results show that the identical-selection strategy performs the worst, which is consistent with our geometric analysis: in this case, for arbitrary auxiliary points $n_\ell, n_q$ and positions $p_i, p_j$, the planes $P_{\{p_i,p_j,n_\ell\}}$ and $P_{\{p_i,p_j,n_q\}}$, determined by $(p_i, p_j, n_\ell)$ and $(p_i, p_j, n_q)$ respectively, always satisfy

$$P_{\{p_i,p_j,n_\ell\}} = P_{\{p_i,p_j,n_q\}}. \tag{14}$$

As a result, different subspaces are not distinguished and the positional information becomes redundant. Random, grid-based, and learnable selection yield comparable performance. This indicates that as long as the auxiliary points satisfy a basic non-degeneracy condition, SpRePE is quite robust to the specific choice of auxiliary-point selection scheme.

Table 6: Auto-regressive forecasting results on the ERA5 dataset using different auxiliary-point selection strategies. Performance is evaluated by Anomaly Correlation Coefficient (ACC, higher is better). We measure ACC at the 4th(+24h), 20th (+120h) and 40th (+240h) steps.

| PE scheme | ACC at 24h ↑ | | | ACC at +120h ↑ | | | ACC at +240h ↑ | | |
|---|---|---|---|---|---|---|---|---|---|
| | $Z500$ | $T2M$ | $T850$ | $Z500$ | $T2M$ | $T850$ | $Z500$ | $T2M$ | $T850$ |
| Random | 0.9962 | 0.9668 | 0.9793 | 0.9047 | 0.8738 | 0.8718 | 0.6181 | 0.7629 | 0.6591 |
| Grid | 0.9965 | 0.9671 | 0.9794 | 0.9095 | 0.8786 | 0.8767 | 0.6155 | 0.7674 | 0.6623 |
| Learnable | 0.9964 | 0.9661 | 0.9788 | 0.9089 | 0.8722 | 0.8736 | 0.6144 | 0.7557 | 0.6587 |
| Identical | 0.9962 | 0.9655 | 0.9782 | 0.8943 | 0.8642 | 0.8621 | 0.6050 | 0.7510 | 0.6475 |

Table 7: Auto-regressive forecasting results on the ERA5 dataset using different masking strategies. Performance is evaluated by Anomaly Correlation Coefficient (ACC, higher is better). We measure ACC at the 4th(+24h), 20th (+120h) and 40th (+240h) steps.

| PE scheme | ACC at 24h ↑ | | | ACC at +120h ↑ | | | ACC at +240h ↑ | | |
|---|---|---|---|---|---|---|---|---|---|
| | $Z500$ | $T2M$ | $T850$ | $Z500$ | $T2M$ | $T850$ | $Z500$ | $T2M$ | $T850$ |
| Fixed-ratio | 0.9965 | 0.9758 | 0.9813 | 0.9121 | 0.9230 | 0.8886 | 0.6333 | 0.8258 | 0.6783 |
| Density-adaptive | 0.9964 | 0.9754 | 0.9809 | 0.9110 | 0.9226 | 0.8882 | 0.6237 | 0.8168 | 0.6705 |
| Learnable | 0.9958 | 0.9751 | 0.9800 | 0.8979 | 0.9159 | 0.8769 | 0.6148 | 0.8204 | 0.6673 |

**Masking Strategy**  Beyond the two masking schemes considered in the main text (fixed-ratio and density-adaptive masks), we also evaluate a learnable masking strategy. Concretely, we introduce a learnable parameter tensor $\tilde{M} \in \mathbb{R}^{1 \times 1 \times N \times d}$ and apply a sigmoid gate, so that each token $i$ can learn its own soft masking pattern over the $d$ elements. All experiments are conducted on ERA5 using the same backbone and training setup as in the main weather-forecasting experiments. The results are presented in Table 7.

Empirically, the learnable mask achieves performance comparable to, and in some cases slightly worse than, the fixed-ratio and density-adaptive masks. Among all variants, the fixed-ratio mask attains the best overall ACC while remaining the simplest and parameter-free. For this reason, we use the fixed-ratio configuration as our primary setting in the experiments, and include the density-adaptive and learnable-mask variants as ablations.

