# OpenReview forum: "SpRePE: A Spherical Geometry-Aware Position Embedding for Vision Transformers"
_ICLR.cc/2026/Conference — Submitted to ICLR 2026_

### Official Review · Reviewer_G8kz · 2025-10-28

**Soundness:** 3
**Presentation:** 2
**Contribution:** 3
**Rating:** 4
**Confidence:** 3

**Summary:**

SpRePE introduces a spherical position embedding for Vision Transformers using Householder reflection matrices to encode positions on a sphere, applied directly to query/key vectors with RoPE-like computational efficiency. The method shows improvements over older standard position embeddings (APE, RPE, ALiBi, RoPE) on Spherical MNIST classification and ERA5 weather forecasting tasks.

**Strengths:**

Principled geometric foundation, using Householder reflections to respect spherical topology while maintaining the computational efficiency and drop-in simplicity of methods like RoPE.

**Weaknesses:**

- Experiments use relatively small-scale tasks (Spherical MNIST, downsampled ERA5 at 128×256 resolution) and lack comparison to specialized spherical architectures or more recent baselines designed for spherical data.
- The improvements over simpler baselines like RoPE are often marginal (e.g., 0.88pp on MNIST), making it unclear whether the added geometric complexity translates to meaningful real-world benefits that justify adoption.
- The paper lacks ablation studies on key design choices (auxiliary point selection, masking strategies), theoretical analysis of when spherical geometry matters most, and evaluation on diverse spherical tasks beyond weather forecasting and toy image classification.

**Questions:**

- When does spherical geometry actually matter?
- How should auxiliary points be selected, and how much do results depend on masking strategies, the number of 3D subspaces, or other hyperparameters that lack principled guidance?
- How does SpRePE perform at production-scale resolutions, on other spherical domains (astronomy, geology), with different architectures (Swin, hierarchical transformers), and does the computational overhead remain negligible at scale?

---

> ### Author Response · Authors · 2025-12-03
> **Response to Reviewer G8kz (1/3)**
>
> Thank you for your feedback and valuable comments. We have addressed each of your questions and concerns in turn and revised the manuscript accordingly to improve its rigor and clarity. We believe these changes have significantly strengthened the paper and made the presentation more complete.
>
> ------
>
> ### Q1. When does spherical geometry actually matter?
>
> Thank you for raising this important question. In our setting, spherical geometry plays a particularly critical role in at least two situations:
>
> 1. **Regions where ERP distortion is strongest.**
>     In equirectangular projection (ERP), points near the poles and near the $0^\circ/360^\circ$ longitude line suffer from polar singularities and longitudinal periodicity. In these regions, spherical effects are much more pronounced than near the equator. Our new attention visualizations (Figure~5) show that SpRePE better captures these geometric effects: for example, it correctly attends to wrap-around neighbours across the image boundary and produces concentrated attention around the poles, in contrast to RoPE’s stripe-like patterns.
> 2. **Long-range autoregressive forecasting on the sphere.**
>     As discussed in prior work (e.g., Bonev et al. (2023, "Spherical Fourier Neural Operators: Learning Stable Dynamics on the Sphere", arXiv:2306.03838)) and reflected in our experiments, respecting spherical geometry becomes increasingly important for long-horizon autoregressive prediction, where small inconsistencies can accumulate over time. In Table\~4, as the autoregressive forecast horizon increases, SpRePE gradually outperforms APE and RoPE.
> Furthermore, Figure\~3 shows that at high latitudes (near the poles), SpRePE yields noticeably lower error than RoPE for long-term forecasts.

---

> ### Author Response · Authors · 2025-12-03
> **Response to Reviewer G8kz (2/3)**
>
> ### Q2. Auxiliary points, masking strategies, and number of 3D subspaces
>
> Thank you for pointing out that our original discussion was not sufficiently detailed. We address each aspect below.
>
> **Auxiliary-point selection.**
>  In the revision, we added an ablation study on the auxiliary-point selection scheme under the same backbone on ERA5. We compare four strategies:
>
> - Random selection: In each layer, auxiliary points are sampled uniformly at random on the sphere and kept fixed;
> - Grid-based selection: In each layer, auxiliary points are chosen from a fixed latitude–longitude grid and kept fixed;
> - Learnable selection: In each layer, auxiliary points are initialized randomly and optimized jointly with the model parameters;
> - Identical selection: In each layer, all auxiliary points coincide at the same location.
>
> The results are shown in the table below, and we have also included them in the revised paper as Table~6.
>
> | SpRePE: Auxiliary-point selection | ACC@24h Z500 | ACC@24h T2M | ACC@24h T850 | ACC@120h Z500 | ACC@120h T2M | ACC@120h T850 | ACC@240h Z500 | ACC@240h T2M | ACC@240h T850 |
> | --------------------------------- | ------------ | ----------- | ------------ | ------------- | ------------ | ------------- | ------------- | ------------ | ------------- |
> | Ramdom Selection| 0.9962| 0.9668| 0.9793| 0.9047| 0.8738| 0.8718| 0.6181| 0.7629| 0.6591|
> | Grid-based Selection| 0.9965| 0.9671| 0.9794| 0.9095| 0.8786| 0.8767| 0.6155| 0.7674| 0.6623|
> | Learnable Selection| 0.9964| 0.9661| 0.9788| 0.9089| 0.8722| 0.8736| 0.6144| 0.7557| 0.6587|
> | Identical Selection| 0.9962| 0.9655| 0.9782| 0.8943| 0.8642| 0.8621| 0.6050| 0.7510| 0.6475|
>
> The identical-selection strategy performs the worst, which is consistent with our geometric analysis: in this case, for any $n_\ell, n_q$ we always have
> $$
> P_{\\{p_i,p_j,n_\ell\\}} = P_{\\{p_i,p_j,n_q\\}},
> $$
> so different subspaces are not distinguished and the positional information becomes redundant. Random, grid-based, and learnable selection yield comparable performance. This indicates that as long as the auxiliary points satisfy a basic non-degeneracy condition, SpRePE is quite robust to the specific choice of auxiliary-point selection scheme.
>
> **Masking strategies.**
>
> In the main paper we study two simple, parameter-free masking schemes for applying SpRePE to $Q$ and $K$: a fixed-ratio mask (“F Mask”) and a density-adaptive mask. Both operate element-wise on $Q,K \in \mathbb{R}^{B \times H \times N \times d}$ via a binary tensor $M \in \\{0,1\\}^{1 \times 1 \times N \times d}$, so that only a subset of the $d$ elements of each token is replaced by its SpRePE-encoded counterpart.
>
> To further assess how much the results depend on the masking strategy, we additionally perform an ablation with a *learnable* mask of the same shape $M$. Concretely, we introduce a learnable parameter tensor $\tilde{M} \in \mathbb{R}^{1 \times 1 \times N \times d}$and use a sigmoid gating,
> $$
> M = \sigma(\tilde{M}),
> $$
> so that each token $i$ can learn its own soft masking pattern over the $d$ elements. These experiments are conducted on the ERA5 dataset using the same backbone and training setup as in the main weather-forecasting experiment.  The results are shown in the table below, and we have also included them in the revised paper in Appendix F (Table~7).
>
> | Masking strategy      | ACC@24h Z500 | ACC@24h T2M | ACC@24h T850 | ACC@120h Z500 | ACC@120h T2M | ACC@120h T850 | ACC@240h Z500 | ACC@240h T2M | ACC@240h T850 |
> | --------------------- | ------------ | ----------- | ------------ | ------------- | ------------ | ------------- | ------------- | ------------ | ------------- |
> | Fix-ratio| 0.9965| 0.9758| 0.9813| 0.9121| 0.9230| 0.8886| 0.6333| 0.8258| 0.6783|
> | Density-adaptive | 0.9964| 0.9754| 0.9809| 0.9110| 0.9226| 0.8882| 0.6237| 0.8168| 0.6705|
> | Learnable| 0.9958| 0.9751| 0.9800| 0.8979| 0.9159| 0.8769| 0.6148| 0.8204| 0.6673|
>
> Empirically, the learnable mask achieves performance comparable to, and in some cases slightly worse than, the fixed-ratio and density-adaptive masks. Among all variants, the fixed-ratio mask attains the best overall ACC while remaining the simplest and parameter-free. For this reason, we use the fixed-ratio configuration as our primary setting in the experiments (its results are already reported in the main text), and include the density-adaptive and learnable-mask variants as ablations.
>
> **Number of 3D subspaces.**
>  In SpRePE, the number of 3D subspaces is *not* a free hyperparameter; it is determined by the transformer’s latent dimension. Concretely, we partition the latent dimension into groups of size 3, so the number of subspaces is $\text{latent dim}/3$, analogous to how RoPE partitions the embedding into 2D rotation blocks.

---

> ### Author Response · Authors · 2025-12-03
> **Response to Reviewer G8kz (3/3)**
>
> ### Q3. Scaling to larger resolutions, architectures, and tasks; computational overhead
>
> We appreciate your question about scaling and practical overhead. Due to time and budget constraints, we were not able to include additional production-scale experiments  within the rebuttal period. We agree that such evaluations would be valuable future work.
>
> That said, we can characterize the computational cost of SpRePE more precisely. As summarized in Table~2:
>
> - The **space complexity** of SpRePE is $\mathcal{O}(N d L)$,
> - The **time complexity** is $\mathcal{O}(B N d L)$,
>
> where $B$ is the batch size, $N$ is the sequence length, $d$ is the embedding dimension, and $L$ is the number of attention layer. The SpRePE encoding is implemented as element-wise operations on $Q$ and $K$ (per-position multiplications) and does not introduce additional matrix multiplications. Consequently, the overhead grows linearly with $N$, $d$, and $L$, and remains small compared to the cost of the attention matrix computation itself—similar to RoPE.
>
> We expect this behaviour to persist at larger scales, although a full production-scale study across domains and architectures is beyond the scope of the present paper and is left for future work.

---

### Official Review · Reviewer_1Wsb · 2025-10-28

**Soundness:** 3
**Presentation:** 2
**Contribution:** 3
**Rating:** 4
**Confidence:** 4

**Summary:**

The paper introduces a positional encoding method for data defined on a sphere. The core concept builds on the RoPe scheme, where positions are encoded using complex exponentials, which are then multiplied (rather than added) to the vectors. This approach is further adapted for datapoints on a 3D-sphere. From what I gathered, the key innovation lies in using Householder reflection matrices (relative to a predefined grid of points, denoted as $n$) to encode the positional data.

**Strengths:**

1. The manuscript is well-written, presenting a novel positional encoding scheme by adapting RoPe for spherical data.

2. The approach is evaluated through benchmarking against several existing methods from the literature.

**Weaknesses:**

1. The manuscript does not provide a more formal statement for the proof in Appendix C. Although Section 3.1 discusses the result, its current form makes it unclear what the formal statement is and what its implications are (see also Question 3).

2. It would be helpful to include one or two additional experimental settings. For instance, the Spherical CNN approach by Cohet et al. (2018) offers experimental settings that could be considered for comparison.

**Questions:**

1. Can you explain how you are choosing the grid of $n$. There is a discussion in section 3.1 about this saying "... for every pair of $(p_1, p_2)$, any two auxiliary points $n_1$ and $n_2$ must define distinct sections.", but how is this condition enforced both for uniform and non-uniform grid?

2. It is not very clear to me the details of "Density-adaptive Mask". Can you elaborate on that?

3. In the derivation provided in Appendix C, what is is the main result and the consequence?

(I have kept my score on the lower end and will change it based on the answers of the above questions.)

4. The RoPe scheme uses complex exponetials, which can be thought of as spherical harmonics in 2D. Have you thought of extending this idea and designing something using spherical harmonics in 3D?

Typo in line 255: "Noticed" -> "Notice".

---

> ### Author Response · Authors · 2025-12-03
> **Response to Reviewer 1Wsb (1/3)**
>
> Thank you very much for your feedback and valuable comments. We have carefully addressed your questions and concerns and revised the manuscript accordingly to improve its rigor and clarity. We believe these changes have significantly strengthened the paper and made the presentation more transparent.
>
> Below we provide detailed responses to each question.
>
> ### Q1. Choice of the grid of auxiliary points $n$
>
> Thank you for raising this question. Before explaining the selection scheme, we find it helpful to briefly restate the role of auxiliary points.
>
> Given $N$ positions to be encoded, $p_k, k \in \{1,2,\dots,N\}$, each auxiliary point $n_\ell$ determines Householder matrices $R_{n_\ell}(p_k)$. In the attention computation, for arbitrary two positions $p_i$ and $p_j$, the inner product between their query and key vectors contains the factor
> $$
> q_i^\top R_{n_\ell}(p_i)^\top R_{n_\ell}(p_j)\\,k_j,
> $$
> where the product
> $$
> R_{n_\ell}(p_i)^\top R_{n_\ell}(p_j)
> $$
> is a rotation matrix with a clear geometric meaning: it is a rotation whose axis is the normal vector of the plane
> $$
> P_{\\{p_i,p_j,n_\ell\\}}
> $$
> determined by the three points $p_i$, $p_j$, and $n_\ell$, and whose rotation angle equals the central angle between $p_i$ and $p_j$ along the circle given by the intersection of $P_{\{p_i,p_j,n_\ell\}}$ with the unit sphere. This is precisely the core conclusion formalized in Appendix C (see also our response to Q3).
>
> Now consider two auxiliary points $n_\ell$ and $n_q$. If they happen to satisfy
> $$
> P_{\\{p_i,p_j,n_\ell\\}} = P_{\\{p_i,p_j,n_q\\}},
> $$
> i.e., the four points $p_i, p_j, n_\ell, n_q$ are coplanar, then we obtain
> $$
> R_{n_\ell}(p_i)^\top R_{n_\ell}(p_j)=R_{n_q}(p_i)^\top R_{n_q}(p_j).
> $$
>
> In that case, the positional information carried by these two three-dimensional subspaces becomes redundant. This is exactly the degeneracy that we aim to avoid when choosing auxiliary points. Apart from avoiding such cases, we do not impose additional constraints on the auxiliary-point selection.
>
> An observation is that, for points sampled in continuous space on the sphere, the event that two independently chosen auxiliary points yield the same plane $P_{\\{p_i,p_j,n\\}}$ has probability zero. Thus, random selection of auxiliary points satisfies the above non-degeneracy condition with probability one, regardless of whether the underlying grid is uniform or non-uniform.
>
> We have added an ablation study on the auxiliary-point selection strategy. Under the same backbone and on the ERA5 dataset, we compare four strategies:
>
> - Random selection,
> - Grid-based selection,
> - Learnable selection.
> - Identical selection (all auxiliary points coincide)
>
> The results are shown in the table below, and we have also included them in the revised paper as Table~6.
>
> | Auxiliary-point selection | ACC@24h Z500 | ACC@24h T2M | ACC@24h T850 | ACC@120h Z500 | ACC@120h T2M | ACC@120h T850 | ACC@240h Z500 | ACC@240h T2M | ACC@240h T850 |
> | --------------------------------- | ------------ | ----------- | ------------ | ------------- | ------------ | ------------- | ------------- | ------------ | ------------- |
> | Ramdom Selection             | 0.9962       | 0.9668      | 0.9793       | 0.9047        | 0.8738       | 0.8718        | 0.6181        | 0.7629       | 0.6591        |
> | Grid-based Selection         | 0.9965       | 0.9671      | 0.9794       | 0.9095        | 0.8786       | 0.8767        | 0.6155        | 0.7674       | 0.6623        |
> | Learnable Selection          | 0.9964       | 0.9661      | 0.9788       | 0.9089        | 0.8722       | 0.8736        | 0.6144        | 0.7557       | 0.6587        |
> | Identical Selection          | 0.9962       | 0.9655      | 0.9782       | 0.8943        | 0.8642       | 0.8621        | 0.605         | 0.751        | 0.6475        |
>
> The identical-selection strategy performs the worst, which is consistent with our geometric analysis: in this case, for any $n_\ell, n_q$ we always have
> $$
> P_{\\{p_i,p_j,n_\ell\\}} = P_{\\{p_i,p_j,n_q\\}},
> $$
> so different subspaces are not distinguished and the positional information becomes redundant. Random, grid-based, and learnable selection yield comparable performance. This indicates that as long as the auxiliary points satisfy a basic non-degeneracy condition, SpRePE is quite robust to the specific choice of auxiliary-point selection scheme.

---

> ### Author Response · Authors · 2025-12-03
> **Response to Reviewer 1Wsb (2/3)**
>
> ### Q2. Clarification of the density-adaptive mask
>
> Thank you for pointing out that the description of the density–adaptive mask was not sufficiently clear. We clarify both its motivation and its precise implementation below, and then report additional ablation results with learnable masks.
>
> Intuitively, when position encoding depends only on a token’s location on the sphere, highly non-uniform sampling may lead to *representation degeneracy*: tokens in densely sampled regions (e.g., near the polar on ERP grids) would receive very similar positional features, which can make the $Q/K$ representations ill-conditioned and reduce the effective rank of attention. To mitigate this effect, we do not apply SpRePE to all channels of $Q$ and $K$, but instead gate it by a binary mask $M∈\{0,1\}^{1×1×N×d}$ along the token and hidden dimension. For $Q,K \in \mathbb{R}^{B\times H\times N\times d}$, we use:
> $$
> Q = (1-M)\circ Q_0 + M\circ \mathrm{SpRePE}(Q_0,p),\qquad K = (1-M)\circ K_0 + M\circ \mathrm{SpRePE}(K_0,p),
> $$
> where $Q_0,K_0$ are the vanilla projections and $p$ denotes token positions. For each token index $i$ (along the $N$ dimension), the slice $M[:,:,i,:]$ contains approximately $\rho_i d$ ones, so only a fraction $\rho_i$ of the $d$ elements of token $I$ are replaced by their SpRePE-encoded counterparts.
>
> In the density–adaptive mask, $\rho_i$ is chosen as a simple function of latitude:
> $$
> \rho_i = \mathrm{clip}\bigl(\rho_{\max} \cos\phi_i,\;\rho_{\min},\;\rho_{\max}\bigr).
> $$
> This yields a basic density-aware masking scheme that takes into account the non-uniform cell areas on the sphere. In parallel, we also evaluate a fixed-ratio (“F Mask”) variant with $\rho_i \equiv \rho$; in our experiments, this fixed-ratio mask performs similarly or slightly better than the density–adaptive mask, and we have also reported its results in the main text.
>
> We further perform an ablation with a learnable mask that has the same shape $M \in \\{0,1\\}^{1 \times 1 \times N \times d}$. Concretely, we introduce a learnable parameter tensor $\tilde{M} \in \mathbb{R}^{1 \times 1 \times N \times d}$ and use a sigmoid gating,
> $$
> M = \sigma(\tilde{M}),
> $$
> so that each token $i$ can learn its own soft masking pattern over the $d$ elements. These experiments are conducted on the ERA5 dataset using the same backbone and training setup as in the main weather-forecasting experiment.  The results are shown in the table below, and we have also included them in the revised paper in Appendix F (Table~7).
>
> | Masking strategy      | ACC@24h Z500 | ACC@24h T2M | ACC@24h T850 | ACC@120h Z500 | ACC@120h T2M | ACC@120h T850 | ACC@240h Z500 | ACC@240h T2M | ACC@240h T850 |
> | --------------------- | ------------ | ----------- | ------------ | ------------- | ------------ | ------------- | ------------- | ------------ | ------------- |
> | Fix-ratio         | 0.9965       | 0.9758      | 0.9813       | 0.9121        | 0.923        | 0.8886        | 0.6333        | 0.8258       | 0.6783        |
> | Density-adaptive  | 0.9964       | 0.9754      | 0.9809       | 0.9110        | 0.9226       | 0.8882        | 0.6237        | 0.8168       | 0.6705        |
> |Learnable         | 0.9958       | 0.9751      | 0.9800       | 0.8979        | 0.9159       | 0.8769        | 0.6148        | 0.8204       | 0.6673        |
>
> Empirically, the learnable mask achieves performance comparable to, and in some metrics slightly worse than, the fixed-ratio and density–adaptive masks. In our experiments, the fixed-ratio mask is simple, parameter-free, and achieves the best overall performance; we present its results together with those of the density-adaptive mask in the main paper for comparison.

---

> ### Author Response · Authors · 2025-12-03
> **Response to reviewer 1Wsb (3/3)**
>
> ### Q3. Main result and consequence of Appendix C
>
> Thank you for asking us to clarify the main message of Appendix C. In the original version, this appendix was presented merely as a supplementary proof, and we did not explicitly articulate how its conclusion connects to the main body of the paper; in the revision, we now make this connection explicit.
>
> The main result of Appendix C is to show that the encoding produced by SpRePE has a precise geometric interpretation. Specifically, we prove that
> $$
> R_{n_\ell}(p_i)^\top R_{n_\ell}(p_j)
> $$
> is a rotation whose axis is the normal vector of the plane $P_{\\{p_i,p_j,n_\ell\\}}$ determined by $p_i$, $p_j$, and $n_\ell$, and whose rotation angle equals the central angle between $p_i$ and $p_j$ along the circle $P_{\\{p_i,p_j,n_\ell\\}} \cap \mathbb{S}^2$.
>
> The consequence is twofold:
>
> 1. Each auxiliary point defines a three-dimensional subspace in which the relative position between $p_i$ and $p_j$ is encoded as a geometrically interpretable 3D rotation.
> 2. This geometric characterization directly motivates our rule for choosing auxiliary points (see Q1): if two auxiliary points yield the same plane $P_{\\{p_i,p_j,n\\}}$, then the corresponding rotations coincide and the positional information in those two subspaces is redundant, which we want to avoid.
>
> In the revised Appendix C, we now explicitly state this result, then present the derivation and summarize its implications for the design of SpRePE. We hope this makes the purpose and consequence of Appendix C much clearer.
>
> ### Q4. Relation to RoPE and possible extension via 3D spherical harmonics
>
> You are correct that the RoPe scheme uses complex exponentials, and it can be viewed as exploiting the relation between $e^{i\theta}$ and 2D rotation matrices.
>
> During the development of SpRePE, we did consider using 3D rotation matrices to encode positions on the sphere. However, we found that such designs typically have less transparent geometry in the context of attention. For instance, even if we introduce auxiliary points, the rotation matrix that maps $p_i$ to $n_\ell$ in $\mathbb{R}^3$ is not unique, and the product of two 3D rotations (which is what appears inside attention) interacts in a relatively complicated way. This makes it difficult to obtain a clean geometric interpretation analogous to Appendix C.
>
> By contrast, Householder reflection matrices admit a simple and unique construction from $(p_k,n_\ell)$, and they lead directly to the clear geometric meaning discussed above. For this reason, we decided to build SpRePE on reflection matrices rather than on general 3D rotations.
>
> We agree that designing positional encodings directly from 3D spherical harmonics is a very interesting research direction. In our view, an important first step would be to establish a principled correspondence between spherical-harmonic representations and geometric quantities that are most relevant for attention (such as relative rotations or geodesic distances). This investigation is somewhat beyond the scope of the present work, which focuses on introducing and analyzing SpRePE, but we consider it a promising direction for future work.

---

### Official Review · Reviewer_NJ9S · 2025-10-31

**Soundness:** 2
**Presentation:** 2
**Contribution:** 2
**Rating:** 4
**Confidence:** 3

**Summary:**

This paper proposes Spherical Reflection Position Embedding (SpRePE), a geometry-aware and lightweight position embedding method for Transformers operating on spherical data. The approach leverages the Householder matrix to encode absolute positions on the sphere and incorporates explicit relative position dependencies directly into the attention formulation. Unlike previous spherical embedding methods that rely on additional modules or complex architectures, SpRePE maintains high computational efficiency with minimal parameter overhead. Experiments are conducted on spherical image classification and global weather forecasting tasks, showing improvements over existing baselines such as APE, RPE, ALiBi, and RoPE.

**Strengths:**

The paper targets an important and under-explored area — transformer modeling on spherical domains, which has strong relevance to scientific applications like meteorology and astronomy.

The formulation is elegant, providing a theoretically grounded yet efficient way to encode positional information on non-Euclidean manifolds.

The method is lightweight, requiring minimal architectural changes and extra parameters.

The paper is well written, and the idea is clearly explained with sound mathematical motivation.

**Weaknesses:**

The performance gain is quite small compared to APE — for example, in Table 3, accuracy only improves from 96.29 to 96.74, and in Table 4, from 0.9954 to 0.9965. Such marginal improvements raise concerns about the practical significance of the proposed method.

The evaluation is limited to a few specific datasets (spherical image and weather data). To better demonstrate generalization and robustness, it would be valuable to test on more widely used image or language datasets (e.g., ImageNet, COCO, or text benchmarks).

It is unclear whether SpRePE provides benefits beyond specialized spherical data — this limits its broader impact and may restrict its applicability.

**Questions:**

see weakness

---

> ### Author Response · Authors · 2025-12-03
> **Response to Reviewer NJ9S (1/2)**
>
> Thank you for your careful reading of our manuscript and for the constructive feedback. We appreciate your positive comments on the importance of the problem, the elegance of the formulation, and the lightweight nature of our method. Below we address your concerns point by point and clarify the intended scope and impact of SpRePE.
>
> ### W1. Magnitude of performance gains compared to APE
>
> We would like to clarify the following points:
>
> **Strong baselines.**
>  The baselines we compare against (APE, RoPE, RPE, etc.) are all well-established  methods. On the benchmarks we consider, these schemes are the mainstream choices for transformer models. The fact that SpRePE achieves consistent improvements over them while introducing almost no additional learnable modules or complex architectural modifications is non-trivial.
>
> **Geometry becomes more important in challenging regimes.**
>  SpRePE is specifically designed for settings where spherical geometry plays a critical role, in particular:
>
> - In equirectangular projections (ERP), spherical distortion is much stronger near the poles than near the equator;
> - In long-range autoregressive forecasting, spherical inconsistencies tend to accumulate over time. (Bonev et al. (2023, "Spherical Fourier Neural Operators: Learning Stable Dynamics on the Sphere", arXiv:2306.03838))
>
> In Table 4, as the autoregressive prediction horizon increases, SpRePE gradually outperforms APE and RoPE, which is consistent with this intuition. In addition, Figure 3 shows that in high-latitude regions (near the poles), SpRePE clearly outperforms RoPE. These observations indicate that the geometric design of SpRePE brings tangible performance gains precisely in scenarios where spherical effects are most pronounced.
>
> **Novelty and interpretability of SpRePE.**
> Beyond the numerical results, we believe the main contribution of SpRePE lies in proposing a geometry-aware positional encoding: on the one hand, it explicitly encodes relative positions on the sphere via Householder reflections; on the other hand, the encoding admits a clear geometric interpretation, which is formalized in Appendix C. Such a theoretically grounded and interpretable design that can match or surpass strong baselines with almost no additional overhead already demonstrates the practical value of the method.
>
> We hope the above clarifications explain why we view these improvements as meaningful in the context of strong baselines and geometrically challenging tasks.
>
> ### W2. Evaluation on a limited set of datasets
>
> We fully agree that adding more experiments on large-scale benchmarks would further strengthen the empirical part of the paper. Due to limitations in computational resources and time during the submission and rebuttal periods, we were unfortunately not able to include additional large-scale experiments beyond the current results. Instead, we chose to focus on two representative and structurally distinct spherical tasks:
>
> - **Spherical image classification**, which is closer to vision applications such as omnidirectional cameras and panoramic images;
> - **Global weather forecasting**, which is a typical task for geophysical and scientific modeling on the sphere.
>
> These two tasks differ substantially in modality (static images vs. spatio-temporal fields) and task objective (classification vs. autoregressive forecasting), yet SpRePE consistently outperforms the baselines in both cases. We believe this already provides evidence of the robustness of our method to different types of spherical data.
>
> At the same time, we view applying SpRePE (or its extensions) to more conventional 2D image or language benchmarks as a very appealing direction for future work—for example, in settings where data can be naturally mapped to the sphere (such as 360° vision) or where spherical parameterizations offer potential advantages. However, a systematic exploration of these directions would require substantial additional engineering effort and computational resources, and thus lies beyond the scope of the current submission, whose main goal is to focus on transformer-based modeling on spherical domains.

---

> ### Author Response · Authors · 2025-12-03
> **Response to Reviewer NJ9S (2/2)**
>
> ### W3. Applicability beyond “specialized spherical data”
>
> You also express concern that it is currently unclear whether SpRePE can provide benefits beyond “specialized spherical data”, which might limit its broader impact.
>
> Here we would like to emphasize that the intended design scope of SpRePE is precisely transformer-based modeling on spherical domains. While this may appear “specialized” from the perspective of standard ML benchmarks, spherical data are in fact very common in many important scientific and industrial applications, including:
>
> - numerical weather prediction and climate modeling;
> - geophysics and Earth observation;
> - omnidirectional and panoramic imaging;
>
> In these applications, spherical geometry is not a minor detail that can be ignored, but a fundamental structural property of the data. Existing positional encodings (APE, RoPE, etc.) were primarily designed for Euclidean grids and do not explicitly exploit this geometric structure. SpRePE is specifically proposed to fill this gap by providing a lightweight, geometry-aware positional encoding that can be directly integrated into standard transformer architectures with only minimal architectural changes.
>
> We agree that extending the core ideas of SpRePE to more general manifolds is a very promising research direction. We believe similar ideas could inspire positional encodings tailored to other settings, such as the scenario discussed in Question 3 of reviewer W8Az. However, a systematic study of such generalized extensions would require substantial additional work and is beyond the scope of this paper; we therefore leave these directions to future work.

---

### Official Review · Reviewer_W8Az · 2025-10-31

**Soundness:** 3
**Presentation:** 3
**Contribution:** 3
**Rating:** 6
**Confidence:** 3

**Summary:**

This paper proposes SpRePE (Spherical Reflection Position Embedding), a geometry-aware and efficient positional encoding for ViTs on spherical data. It uses Householder reflections to encode absolute positions on the sphere and lets relative information emerge via attention inner products. No extra trainable parameters, and it’s drop-in to standard Transformers. Validated on Spherical MNIST and ERA5.

**Strengths:**

1.	Novel geometric formulation: Reflection-based, sphere-aware encoding; handles poles & longitudinal wrap-around better than planar PE.
2.	Minimal overhead: No architecture change; same complexity class as RoPE; avoids quadratic RPE.
3.	Empirical gains: Better accuracy/robustness (especially at high latitudes & long horizons).
4.	Comprehensive eval: Comparisons + ablations (masking, robustness).
5.	Clarity & reproducibility: Derivations are clean; code & settings well documented.

**Weaknesses:**

1.	Dataset breadth: Only Spherical MNIST & ERA5; lacks panoramic CV / point-cloud / remote-sensing detection tasks.
2.	Qualitative insight: Could add attention maps / distance heatmaps to visualize geometric effects.
3.	Ablation depth: Need to isolate the contribution of the geometric term vs. reflection itself; clarify sensitivity to auxiliary points {n_i}.
4.	Theory rigor: Proof that reflections yield correct relative encoding could be strengthened.
5.	Baselines: Missing comparisons with newer geometry-aware methods (e.g., Sphere2Vec, Heal-Swin) on same backbones.

**Questions:**

1.	Auxiliary points: How are {n_i} chosen (fixed grid / learned / random)? Sensitivity & stability?
2.	Masking: Tried learned or entropy-gated masks instead of cosine-latitude heuristics?
3.	Generality: Extendable to other manifolds (hyperbolic / cylindrical)? What changes?
4.	Scale: Results on larger backbones / higher-res ERP (e.g., ViT-L/16 @ 1024×2048)? Memory trade-offs?
5.	Interpretability: Any visualization of reflection effects in latent space or attention geometry?

---

> ### Author Response · Authors · 2025-12-03
> **Response to Reviewer W8Az (1/2)**
>
> Thank you for your feedback and valuable comments. We have addressed each of your questions individually and revised the manuscript accordingly to strengthen its rigor and clarity. We believe these changes have substantially improved the paper and made the presentation clearer and more complete.
>
> ### Q1 Auxilary points
>
> Thank you for pointing out the unclear part regarding the choice of  auxiliary points. As you noted, the selection strategy does influence the quality of the encoding. In the revised version, we added an ablation study on the auxiliary-point selection scheme. Under the same backbone on ERA5, we compare four strategies:
>
> - Random selection: In each layer, auxiliary points are sampled uniformly at random on the sphere and kept fixed;
> - Grid-based selection: In each layer, auxiliary points are chosen from a fixed latitude–longitude grid and kept fixed;
> - Learnable selection: In each layer, auxiliary points are initialized randomly and optimized jointly with the model parameters;
> - Identical selection: In each layer, all auxiliary points coincide at the same location.
>
> The results are shown in the table below, and we have also included them in the revised paper in Appendix F (Table~6).
>
> | Auxiliary-point selection | ACC@24h Z500 | ACC@24h T2M | ACC@24h T850 | ACC@120h Z500 | ACC@120h T2M | ACC@120h T850 | ACC@240h Z500 | ACC@240h T2M | ACC@240h T850 |
> | --------------------------------- | ------------ | ----------- | ------------ | ------------- | ------------ | ------------- | ------------- | ------------ | ------------- |
> | Ramdom Selection             | 0.9962       | 0.9668      | 0.9793       | 0.9047        | 0.8738       | 0.8718        | 0.6181        | 0.7629       | 0.6591        |
> | Grid-based Selection        | 0.9965       | 0.9671      | 0.9794       | 0.9095        | 0.8786       | 0.8767        | 0.6155        | 0.7674       | 0.6623        |
> | Learnable Selection          | 0.9964       | 0.9661      | 0.9788       | 0.9089        | 0.8722       | 0.8736        | 0.6144        | 0.7557       | 0.6587        |
> | Identical Selection          | 0.9962       | 0.9655      | 0.9782       | 0.8943        | 0.8642       | 0.8621        | 0.6050         | 0.7510        | 0.6475        |
>
> We observe that the identical-selection strategy performs the worst, which is consistent with our geometric analysis: in this case, for any $n_\ell, n_q$ we always have
> $$
> P_{\\{p_i,p_j,n_\ell\\}} = P_{\\{p_i,p_j,n_q\\}},
> $$
> so different subspaces are not distinguished and the positional information becomes redundant. Random, grid-based, and learnable selection yield comparable performance. This indicates that as long as the auxiliary points satisfy a basic non-degeneracy condition, SpRePE is quite robust to the specific choice of auxiliary-point selection scheme.
>
> ### Q2 Masking
>
> We appreciate the suggestion to explore alternative masking schemes. In the main paper we evaluate two simple, parameter–free strategies (fixed–ratio and density–adaptive masks). Following your advice, we have now added an ablation with a *learnable* mask of shape $M \in \mathbb{R}^{1 \times 1 \times N \times d}$, implemented as a sigmoid gate applied element-wise to \(Q\) and \(K\). The new experiments are conducted on ERA5 using the same backbone and training setup as in the main weather–forecasting results. The results are shown in the table below, and we have also included them in the revised paper in Appendix F (Table~7).
>
> | Masking strategy | ACC@24h Z500 | ACC@24h T2M | ACC@24h T850 | ACC@120h Z500 | ACC@120h T2M | ACC@120h T850 | ACC@240h Z500 | ACC@240h T2M | ACC@240h T850 |
> | --------------------------------- | ------------ | ----------- | ------------ | ------------- | ------------ | ------------- | ------------- | ------------ | ------------- |
> | Fix-ratio                   | 0.9965       | 0.9758      | 0.9813       | 0.9121        | 0.9230       | 0.8886        | 0.6333        | 0.8258       | 0.6783        |
> | Density-adaptive             | 0.9964       | 0.9754      | 0.9809       | 0.9110        | 0.9226       | 0.8882        | 0.6237        | 0.8168       | 0.6705        |
> | Learnable                    | 0.9958       | 0.9751      | 0.9800       | 0.8979        | 0.9159       | 0.8769        | 0.6148        | 0.8204       | 0.6673        |
>
> Empirically, the learnable mask achieves performance comparable to, and in some cases slightly worse than, the fixed-ratio and cosine-latitude heuristic masks. Among all variants, the fixed-ratio mask attains the best overall ACC while remaining the simplest and parameter-free. For this reason, we use the fixed-ratio configuration as our primary setting in the experiments (its results are already reported in the main text), and include the density-adaptive and learnable-mask variants as ablations.

---

> ### Author Response · Authors · 2025-12-03
> **Response to Reviewer W8Az (2/2)**
>
> ### Q3 Generality
>
> The question on *generality* is indeed very interesting. While the current formulation of SpRePE cannot be directly applied to hyperbolic or cylindrical geometries, we believe that the underlying idea is transferable.
>
> For example, a cylindrical domain can be viewed as the Cartesian product of a circle and a one-dimensional Euclidean axis. The circular part can be regarded as a special case of SpRePE where all patches share the same latitude, whereas RoPE is directly applicable to the 1D Cartesian coordinate. In this setting, one could follow an approach similar to 2D RoPE: split the latent space into $2n$ subspaces, apply RoPE to $n$ of them (for the axial direction) and SpRePE to the remaining $n$ (for the circular direction). This hybrid design would preserve the key idea of SpRePE—using reflections to encode non-Euclidean relative positions—while remaining compatible with standard Euclidean encodings.
>
> We emphasize that this is a conceptual extension and we have not implemented or evaluated it in this work; a thorough study of such generalizations is beyond the scope of the current paper and left for future research.
>
> ### Q4 Scale
>
> Due to time and budget constraints, we are not able to provide larger-scale experiments at this stage. However, we can estimate the memory trade-off of SpRePE.
>
> As summarized in Table~2, the space complexity of SpRePE is $\mathcal{O}(N d L)$, where $N$ is the sequence length, $d$ is the embedding dimension, and $L$ is the number of attention layers. In larger models, the memory overhead contributed by SpRePE grows linearly with these quantities, which is of the same order as RoPE. Compared with the overall memory cost of the attention computation itself, this additional term is relatively small, so we do not expect SpRePE to become a practical memory bottleneck.
>
> ### Q5 Interpretablity
>
> As requested, we added an attention visualization on the ERA5 dataset (Figure~5). For each query position, we show three panels: (left) the attention map with RoPE, (middle) the attention map with SpRePE, and (right) the geodesic distance $d(q,k)$ from the query position to every key on the sphere.
>
>
>
> - **Query at the North Pole.**
>
>   In the first row of Figure~5, the query token is placed at the North Pole. The geodesic distance map now exhibits concentric structure around the pole. The RoPE attention shows a strip-like pattern along the top rows, whereas SpRePE yields a high-attention region concentrated around the pole that decays as we move towards lower latitudes, which qualitatively matches the geodesic distance heatmap.
>
> - **Query at the equator–prime-meridian intersection.**
>   In the second row of Figure~5, the query token is located on the equator at the left boundary of the ERP grid. Under RoPE, the attention map forms a nearly continuous horizontal band along the same latitude, largely ignoring the wrap-around adjacency between the left and right boundaries. In contrast, SpRePE produces two localized high-attention regions around the query and its wrap-around neighbour on the opposite side of the image, closely following the pattern of small geodesic distance in the right panel. This shows that SpRePE correctly captures the longitudinal wrap-around.
>
> These visualizations provide an intuitive illustration of the “attention geometry” induced by SpRePE and we believe they help improve the interpretability of our method.

---

### Author Response · Authors · 2025-12-03
**Rebuttal Summary for AC**

Dear Area Chair,

Thank you very much for your time and effort. Below we would like to provide a brief summary of our rebuttal and revisions to facilitate your meta-review.

In the rebuttal, we carefully addressed every question raised by the reviewers and revised the manuscript accordingly.

- **Geometric interpretation and Appendix C (Reviewer 1Wsb).**
  Following Reviewer 1Wsb’s comments, we substantially revised Appendix C to state the main result as a formal proposition and streamlined the proof. We now make explicit the geometric interpretation of SpRePE and how it directly motivates our rule for selecting auxiliary points.
- **Auxiliary-point selection and masking strategies (Reviewers W8Az, 1Wsb, G8kz).**
  In response to the concerns about the choice of auxiliary points and masking strategies, we added ablation studies under the  weather forecasting setting. For auxiliary-point selection, the ablations confirm the expected behavior: as long as the degeneracy condition discussed in the paper is avoided, SpRePE is robust to the concrete selection scheme. For masking, we clarified in the main text how the fixed-ratio and density-adaptive masks are implemented, and added an ablation with a learnable mask. The results show that this learnable variant performs comparable to, and in some cases slightly worse than, the existing schemes.
- **When spherical geometry matters (Reviewers NJ9S, G8kz).**
  To address questions about the practical importance of spherical geometry, we emphasized that spherical effects are most critical (i) in regions with strong ERP distortion (near the poles and along the prime meridian) and (ii) under long-horizon auto-regressive forecasting. The empirical results in Table 4 and Figure 3 support the superiority of SpRePE in scenarios where spherical geometry plays an important role.
- **Extensions beyond the sphere (Reviewers NJ9S, W8Az).**
  Regarding possible extensions of SpRePE to other manifolds, we added a conceptual discussion using the cylindrical case as an example and clarified what would need to change. For the text and standard 2D benchmarks mentioned by Reviewer NJ9S, we stressed that SpRePE itself is specifically designed for spherical data, and our current focus is on spherical domains. Nevertheless, we believe the underlying ideas can inspire analogous designs in other geometries, which we leave for future work.
- **Attention-geometry visualizations (Reviewer W8Az).**
  Following Reviewer W8Az’s request for qualitative insight, we added attention-map visualizations on ERA5 (new Figure 5). These plots show that, compared to RoPE, SpRePE aligns attention with spherical geodesic structure more faithfully, both at the pole and at the equator/prime-meridian intersection.
- **Larger models, datasets, and backbones (Reviewers W8Az, G8kz).**
  Due to time and computational constraints, we were unable to add new experiments on substantially larger models or additional large-scale datasets. However, we analyzed the computational and memory overhead of SpRePE and argued that it remains essentially on par with RoPE, and negligible compared to the overall attention cost, even at larger scales.

We believe that our responses and revisions address the vast majority of the reviewers’ concerns and further strengthen the clarity, rigor, and interpretability of SpRePE.

Thank you once again for your valuable time and consideration.

Best regards,

 *SpRePE Authors*

---

### Meta-Review · Area_Chair_1Hgy · 2025-12-28

**Summary:**

There are four reviews of this paper.

Reviewer W8Az has concerns on the limited datasets in experiments, qualitative insight, ablation depth, rigor of theory and missing baselines in comparison.

Reviewer NJ9S’ concerns are focused on limited performance gain over APE, the limited evaluation datasets and the generalization performance of the method.

Reviewer 1Wsb has concerns on the proof of the formal statement, the experimental settings, and some implementation details.

Reviewer G8kz’s concerns are focused on the relatively small-scale tasks, lack of comparison to specialized spherical architectures and more recent baselines, the marginal improvements over simpler baselines like RoPE, lack of ablation studies on key design choices, etc.

**Reviewer Concerns:**

None of the reviewers provide feedback on the authors’ rebuttal. The AC feels that most of Reviewer W8Az’s concerns are addressed, while some concerns (evaluation datasets) are the inherent issues of the current paper. For Reviewer NJ9S, only part of his/her concerns have been addressed. The issues on the applicability and evaluation datasets remain. For Reviewer 1Wsb, the AC feels that part of the concerns (some implementation details) should have been addressed, but the proof of the formal statement may not be well addressed. For Reviewer G8kz, the AC believes that only part of the concerns (some ablation studies) have been addressed by the authors’ rebuttal, while the critical concerns (small-scale tasks, lack of comparison, marginal improvements) remain.

Actually, it can be seen that many reviewers have common concerns on the performance, applicability and theoretical strictness of this work.

**Reviewer Scores:**

The ACs think that Reviewer W8Az is likely to maintain his/her score of 6. For the other three reviewers, the ACs believe that at most one (Reviewer 1Wsb) may increase the score to 6. Considering the competitiveness of ICLR, it is unfortunate that this paper cannot be accepted.

---

### Decision · Program_Chairs · 2026-01-26

Reject